

# Long non-coding RNA, LINC01614 as a potential biomarker for prognostic prediction in breast cancer

Yaozong Wang[1], Baorong Song[1], Leilei Zhu[2] and Xia Zhang[3]

[1] Department of General Surgery, Hwa Mei Hospital (Ningbo No.2 Hospital), University of Chinese Academy of Sciences, Ningbo, China
[2] Department of Radiotherapy, Shanghai East Hospital, Tongji University, Shanghai, China
[3] Breast Cancer Center, Shanghai East Hospital, Tongji University, Shanghai, China

Corresponding author
Xia Zhang,
zhangxia010203@126.com

## ABSTRACT

**Background:** Dysregulated long non-coding RNAs (lncRNAs) may serve as potential biomarkers of cancers including breast cancer (BRCA). This study aimed to identify lncRNAs with strong prognostic value for BRCA.

**Methods:** LncRNA expression profiles of 929 tissue samples were downloaded from TANRIC database. We performed differential expression analysis between paired BRCA and adjacent normal tissues. Survival analysis was used to identify lncRNAs with prognostic value. Univariate and multivariate Cox regression analyses were performed to confirm the independent prognostic value of potential lncRNAs. Dysregulated signaling pathways associated with lncRNA expression were evaluated using gene set enrichment analysis.

**Results:** We found that a total of 398 lncRNAs were significantly differentially expressed between BRCA and adjacent normal tissues (adjusted $P$ value <= 0.0001 and $|logFC| >= 1$). Additionally, 381 potential lncRNAs were correlated Overall Survival (OS) ($P$ value < 0.05). A total of 48 lncRNAs remained when differentially expressed lncRNAs overlapped with lncRNAs that had prognostic value. Among the 48 lncRNAs, one lncRNA (LINC01614) had stronger prognostic value and was highly expressed in BRCA tissues. LINC01614 expression was validated as an independent prognostic factor using univariate and multivariate analyses. Higher LINC01614 expression was observed in several molecular subgroups including estrogen receptors+, progesterone receptors+ and human epidermal growth factor receptor 2 (HER2)+ subgroup, respectively. Also, BRCA carrying one of four gene mutations had higher expression of LINC01614 including AOAH, CIT, HER2 and ODZ1. Higher expression of LINC01614 was positively correlated with several gene sets including TGF-β1 response, CDH1 signals and cell adhesion pathways.

**Conclusions:** A novel lncRNA LINC01614 was identified as a potential biomarker for prognosis prediction of BRCA. This study emphasized the importance of LINC01614 and further research should be focused on it.

## INTRODUCTION

Although the management of breast cancer (BRCA) has been significantly improved over the decades of years, it is still one of the most common cancers worldwide, especially for women. The morbidity and mortality of BRCA is continuously increasing in recent years especially in Europe countries such as the US (*Anderson, Katki & Rosenberg, 2011*). Based on the statistics from American Cancer Society, an estimated more than 260,000 BRCA cases occurred and more than 41,000 patients died because of BRCA in 2018 (*Siegel, Miller & Jemal, 2018*). There are several common treatments for BRCA including surgical resection, chemotherapy, radiotherapy (*DeVita, 1989*). Recently, immunotherapy is becoming a new strategy for some cancer types including BRCA (*Hadden, 1999*). Since BRCA is a disease with high heterogeneity, the underlying mechanism of BRCA has not been totally elucidated (*Garcia-Closas et al., 2008*). Some BRCA cases were resistant to chemotherapy and these patients generally had worse survival. Recently, several molecules have been identified as biomarkers that can be used to determine response to therapy and prognostic prediction such as estrogen receptors (ERs), progesterone receptors (PRs) and human epidermal growth factor receptor 2 (HER2) (*Colleoni & Montagna, 2012*; *Davies et al., 2011*; *Network, 2012*).

With the development of high throughput technique such as next generation sequencing, the biological function of long non-coding RNAs (lncRNAs) has been increasingly identified (*Core, Waterfall & Lis, 2008*). Indeed, more than 75% of the genome are transcribed into noncoding RNAs including lncRNAs and small RNAs such as tRNAs, rRNAs, miRNAs. Among these, lncRNAs are noncoding transcripts more than 200 nucleotide (nt) in length and are lack of protein-coding abilities (*Huarte, 2015*). According to the Encyclopedia of DNA Elements Project Consortium, more than 120,000 lncRNAs have been annotated (*Jalali, Gandhi & Scaria, 2016*). Recently, dysregulated expression of lncRNAs have been found to be involved with cancer development and progression. Some critical lncRNAs have been identified as biomarkers and prognostic factors such as HOTAIR (*Gupta et al., 2010*), SAMMSON (*Leucci et al., 2016*), MALAT1 (*Sun & Ma, 2019*) and H19 (*Zhang et al., 2016a*). Specifically, HOTAIR could increase cancer invasiveness and metastasis by reprograming chromatin state in a manner dependent on Polycomb repressive complex 2 (*Gupta et al., 2010*). LncRNA SAMMSON was found to interact with p32 and regulate homeostasis and metabolism of mitochondria. This effect could increase mitochondrial targeting and pro-oncogenic function of p32 (*Leucci et al., 2016*). Some other lncRNAs are identified to be involved with cell cycle control, tumor growth, drug response. Since lncRNAs may play important roles in the cancer carcinogenesis, it is necessary to identify potential lncRNAs that may serve as useful biomarkers and prognostic predictors by bioinformatic analysis.

This study was aimed to identify oncogenic lncRNAs with prognostic value in BRCA. LncRNAs were analyzed by correlating with tumorigenesis and Overall Survival (OS). A total of 48 lncRNAs were differentially expressed between BRCA and adjacent normal tissues and also associated with OS. LncRNA LINC01614 had higher expression in BRCA

tissues and had stronger prognostic value than other candidate lncRNAs, thus, was selected for further analyses including correlation with clinical parameters, univariate and multivariate analyses using Cox regression model and identifying dysregulating signaling pathways using gene set enrichment analysis (GSEA).

## MATERIALS AND METHODS

### Clinical data collection

LncRNA expression profile of BRCA was downloaded from TANRIC database (https://ibl.mdanderson.org/tanric/_design/basic/index.html). All of these samples in TANRIC database were from the Cancer Genomic Atlas (TCGA, https://cancergenome.nih.gov/). A total of 929 tissue samples were collected and 105 of them were derived from paired normal tissues. The clinical information of these samples were downloaded from TCGA. A total of 824 of these samples had survival information OS. A total of 12,727 lncRNAs were annotated in the expression profile. The clinical characteristics of these patients including age, gender, TNM stage, ER status, PR status, HER2 status, treatment history, PAM50 subtypes.

### Gene set enrichment analysis

We used GSEA to identify dysregulated gene sets between high- and low-LINC01614 expression groups. GSEA was conducted by a JAVA program (http://software.broadinstitute.org/gsea/index.jsp) based on molecular signature database C2 CP (v6.2) which consisted of 4,762 gene sets. The number of random sample permutations was set as 1,000. Generally, if the majority of members of a given gene set had higher expression accompanied with higher risk score, then gene set would have positive enrichment score and considered as enriched based on the nominal $P$ value ($<0.05$), false discovery rate (FDR) ($<25\%$) and normalized enrichment score (NES $> 1$) (*Subramanian et al., 2005*).

### Statistical analysis

All of the statistical analyses were conducted by R (version 3.5.1). Differential expression analysis between BRCA and adjacent normal tissues was performed using "Limma" R package (*Ritchie et al., 2015*). Two-sample $t$-test was used to test the statistical differences of LINC01614 expression between two groups and ANOVA test for more than two groups (e.g., PAM50 subtypes). Chi-squared test or Fisher's exact test was used to compare the clinical differences between high- and low-LINC01614 expression groups. Kaplan–Meier estimate was used to measure the proportion of patients living for a certain time period after surgery. Kaplan–Meier estimate was efficient to deal with censored data. We visualized Kaplan–Meier estimate using "survival" R package. Survival difference was compared using the log-rank test. Univariate and Multivariate Cox analysis were used to test whether LINC01614 expression was an independent prognostic factor. Significance difference was confirmed as $P$ value was less than 0.05.

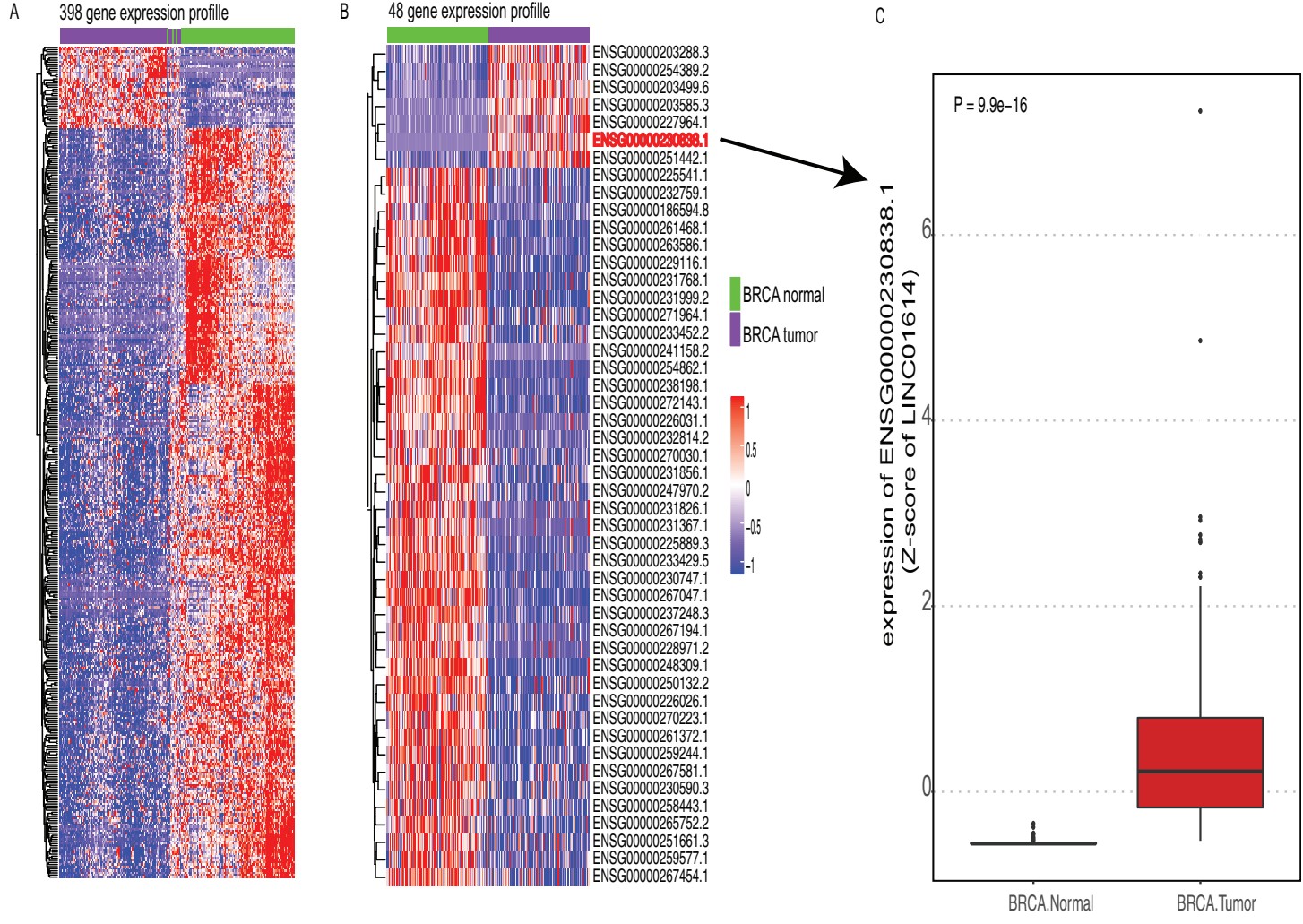

**Figure 1** **Expression profiles of candidate lncRNAs.** (A) Heatmap of 398 lncRNA expression profiles that were differentially expressed between breast cancer (BRCA) and adjacent normal tissues (adjusted *P* value <= 0.0001, |logFC| >= 1). Rows represent lncRNAs and columns represent patients. (B) Heatmap of 48 lncRNA expression profiles that were differentially expressed in (A) with prognostic value (*P* value < 0.05). (C) Boxplot of candidate lncRNA (LINC01614) between BRCA and paired normal tissues.               

## RESULTS

### Identification of candidate lncRNAs with prognostic value in BRCA

The work flow of this study was shown in Fig. S1. According to the TANRIC database, a total of 12,727 lncRNAs were annotated. Firstly, we excluded lncRNAs that could not be detected in at least 80% of these 824 samples. By doing so, 4,537 lncRNAs remained for further analysis. Since paired adjacent normal tissues in 105 of these samples were available, we performed differential expression analysis between these 105 paired samples. By doing so, 398 lncRNAs were differentially expressed between BRCA and adjacent normal tissues (|lgFC| >= 1 and adjusted *P* value <= 0.0001, Fig. 1A). To further screen out lncRNAs with prognostic value, we correlated these 4,537 lncRNAs with OS. A total of 381 lncRNAs were found associated with OS (*P* value < 0.05 and |HR| > 1, Table S1). Subsequently, we identified 48 lncRNAs that were differentially expressed and had

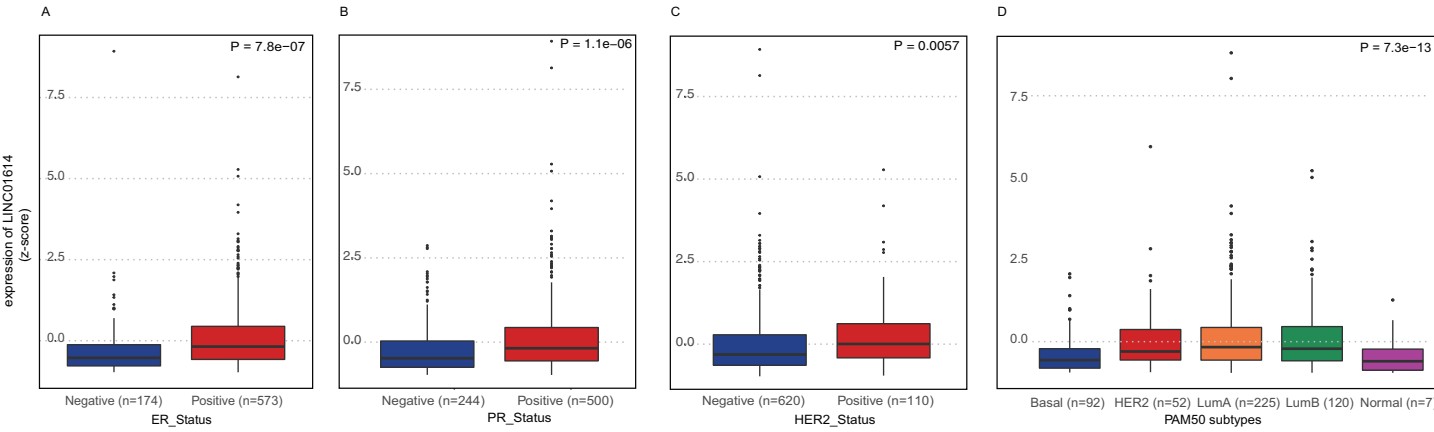

**Figure 2 Expression comparison of LINC01614 among molecular subgroups or subtypes of BRCA.** (A) ER positive vs. ER negative. (B) PR positive vs. PR negative. (C) HER2 positive vs. HER2 negative. (D) PAM50 molecular subtypes.

prognostic value as well (Fig. 1B). Among six lncRNAs that were highly expressed in BRCA tissues, one lncRNA named LINC01614 had stronger prognostic value than others (HR > 1 and *P* value <0.01, Fig. 1C; Table S1). Thus, we selected LINC01614 for further analysis.

## Correlation between LINC01614 expression and clinical characteristics in BRCA

The clinical characteristics of these 824 BRCA patients were shown in Table S2. The age of these patients ranged from 26 to 90 and the median was 58. There were only nine males in this cohort. Stage I patients were found in 140 patients, stage II in 471, stage III in 178 and stage IV in 14. Among these patients, 573 were ER positive, 500 were PR positive and 110 were HER2 positive. Only 65 patients had treatment information and 41 of them were treated. In terms of PAM50 molecular subtypes according to one paper in 2012 (*Network, 2012*), 22 patients were classified as Normal-like subtype, 138 patients were classified as Basal-like subtype, 64 patients were classified as HER2-enriched subtype, 413 patients were classified as Luminal-A subtype and 187 patients were classified as Luminal-B subtype.

These patients were classified into two groups based on the expression of LINC01614 and the median expression was set as the cutoff value. The proportions of ER status ($P < 0.0001$ ), PR status ($P < 0.0001$), HER2 status ($P = 0.0006$) and PAM50 subtypes ($P < 0.0001$) were significantly different between high- and low-expression group as shown in Table S2. Although there was no significant difference between these two groups ($P = 0.076$), there were more stage III and IV cases in high-expression group ($n = 102$) than that in low-expression group ($n = 90$). We further compared the expression differences among different molecular subgroups. Higher expression of LINC01614 was observed in ER positive group ($P < 0.0001$, Fig. 2A), PR positive group ($P < 0.0001$, Fig. 2B), and HER2 positive group ($P = 0.0057$, Fig. 2C). Significant difference was also observed among PAM50 subtypes ($P < 0.0001$, Fig. 2D). Specifically, Luminal A and B had higher expression of LINC01614 followed by HER2, Normal and Basal subtypes.

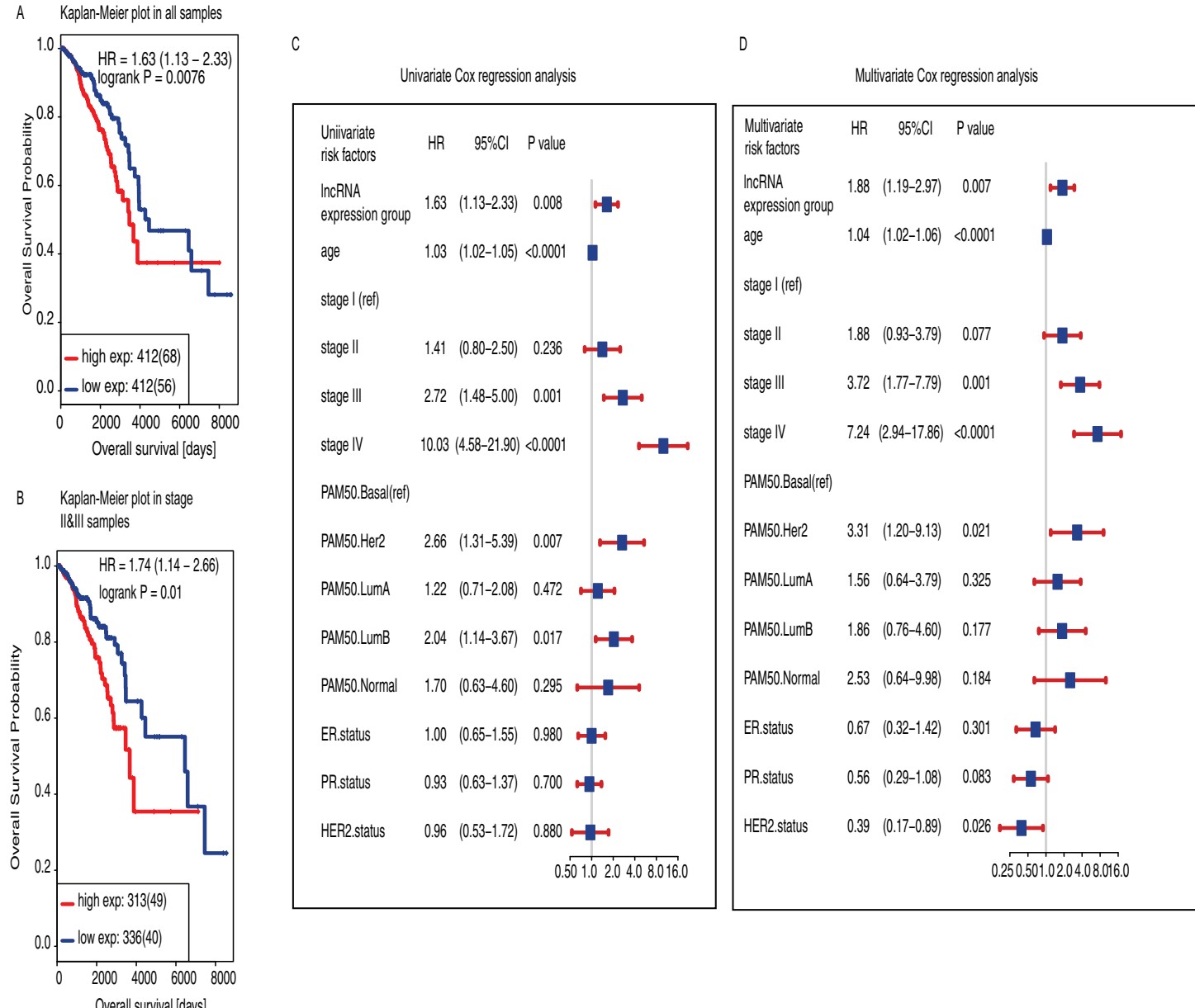

**Figure 3 Survival analysis of LINC01614 based on two groups classified using the median expression of LINC01614.** (A) Kaplan–Meier estimates of Overall Survival (OS) in 824 BRCA samples. (B) Kaplan–Meier estimates of OS stage II and III BRCA samples ($n$ = 647). The tick marks on the Kaplan–Meier curves indicated the censored cases. The differences between the two curves were determined by the two-side log-rank test. (C) Univariate Cox regression proportional hazards regression analysis between clinical features. (D) Multivariate Cox regression proportional hazards regression analysis of LINC0614 expression groups adjusted by other clinical features including age, TNM stage, PAM50, ER, PR and HER2 status. Age was regarded as continuous variable. Gender was not included because there were only nine male subjects and only one in low-expression group. Solid squares indicated the hazard ratio (HR) of death, and close-ended horizontal lines represent the 95% confidence intervals (CI). All $P$ values were calculated using Cox regression hazards analysis.

## Prognostic value of LINC01614 in BRCA

The median follow-up time was 32.5 months and only 15% of these patients were dead ($n$ = 125). Initially, we found that high-expression group had shorter survival than low-expression group ($P$ = 0.0076, HR = 1.63, 95% CI [1.13–2.33], Fig. 3A). We also found

that the prognostic value of LINC01614 was also significant only in TNM stage II and III patients ($P$ = 0.01, HR = 1.71, 95% CI [1.14–2.66], Fig. 3B). We included age, TNM stage, PR status, ER status, HER2 status and PAM50 subtypes for Univariate and Multivariate analysis using Cox regression model. Age was included as a continuous variable in the model while the others were included as categorical variables. We did not include gender in Cox regression model as there were only nine male patients and only one was in low-expression group.

As for univariate Cox regression analysis, we found that high LINC01614 expression was significantly associated with worse survival ($P$ = 0.008, HR = 1.63, 95% CI [1.13–2.33], Fig. 3C) and also for age ($P$ < 0.0001, HR = 1.03, 95% CI [1.02–1.05]), stage III ($P$ = 0.001, HR = 2.72, 95% CI [1.48–5.00]), stage IV ($P$ < 0.0001, HR = 10.03, 95% CI [4.58–21.90]), PAM50 HER2 subtype ($P$ = 0.007, HR = 2.66, 95% CI [1.31–5.39]), PAM50 Luminal B ($P$ = 0.017, HR = 2.04, 95% CI [1.14–3.67]). We then performed multivariate Cox regression analysis and found that LINC01614 expression was an independent prognostic factor for BRCA ($P$ = 0.007, HR = 1.88, 95% CI [1.19–2.97], Fig. 3D) as well as for age ($P$ < 0.0001, HR = 1.04, 95% CI [1.02–1.06]), stage III ($P$ = 0.001, HR = 3.72, 95% CI [1.77–7.79]), stage IV ($P$ < 0.0001, HR = 7.24, 95% CI [2.94–17.86]), PAM50 HER2 subtypes ($P$ = 0.021, HR = 3.31, 95% CI [1.20–9.13]) and HER2 status ($P$ = 0.026, HR = 0.39, 95% CI [0.17–0.89]).

We further performed subgroup analyses based on PR status, ER status and HER2 status (Fig. S2). Our results indicated that prognostic value of LINC01614 could be observed in ER positive group ($P$ = 0.032, HR = 1.62, 95% CI [1.04–2.53]) and HER2 negative group ($P$ = 0.037, HR = 1.56, 95% CI [1.02–2.37]). Although there were no significant differences in other subgroups, we observed the trend that high-expression group had shorter survival than low-expression group among these subgroups. More samples were needed to validate these results.

## Gene mutations correlated with LINC01614 expression

We also explored the correlations between LINC01614 expression and gene mutations status using the inbuilt functions of the TANRIC database. We found that the expression of LINC01614 was significant associated with four gene mutations including AOHA, CIT, HER2 and ODZ1. Consistently, patients carrying any one of these gene mutations had higher expression of LINC01614 ($P$ = 0.0064 for AOHA, $P$ = 0.0004 for CIT, $P$ = 0.0057 for HER2 and $P$ = 0.0011 for ODZ1, Fig. S3).

## Dysregulated pathways associated with LINC01614 expression

We performed GSEA to identify dysregulated gene sets between high- and low-expression group. A total of 799 and 42 gene sets were significantly (FDR < 25%, normal $P$ value < 0.05) enriched in high- and low-expression group, respectively (Table S3). Among these gene sets, some of them were critical in the development and progression of cancer such as TGF-β1 response, CDHI signaling via CTNNB1, integrin3 and integrin cell surface interaction pathways. High-expression group was also enriched with one gene signature associated with ductal carcinoma. Finally, we also found that genes associated with Estradiol treatment were enriched in high-expression group (Figs. 4A–4F).

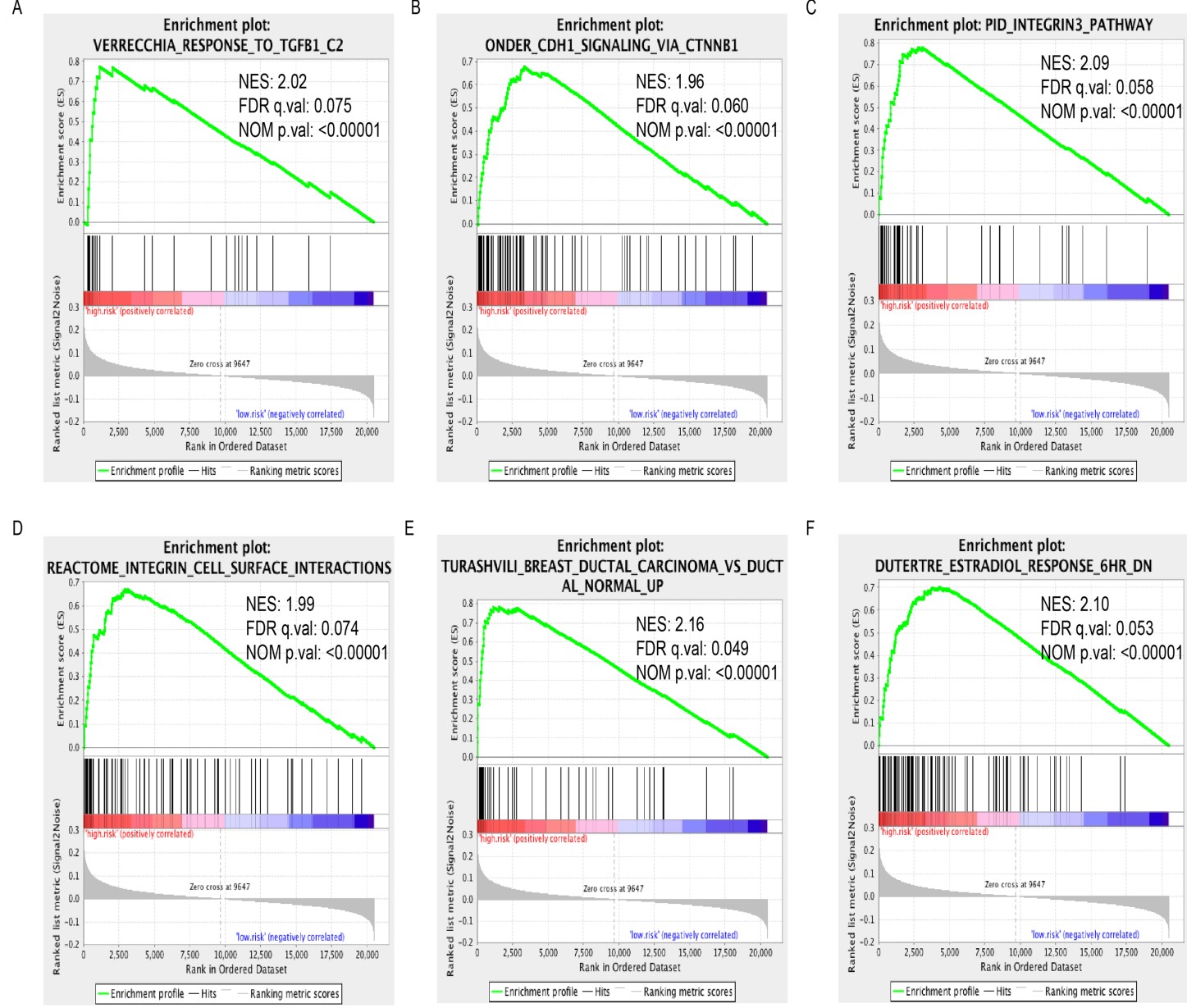

**Figure 4 Enrichment plots of GSEA based on LINC01614 expression group.** GSEA indicated that TGF-β1 response (A), CDHI signaling via CTNNB1 (B), integrin3 (C) and integrin cell surface interaction (D), ductal carcinoma signature (E) and Estradiol treatment response signature were significantly enriched in high-expression group (F). Normalized enrichment score (NES); false discovery rate (FDR).

# DISCUSSION

Taking the advantage of high-throughput sequencing technologies, increasingly comprehensive transcriptomes have been recognized and identified. The roles of lncRNAs have been investigated and emphasized (*Huarte, 2015*). It has been found that lncRNAs play important roles in multiple biological processes including chromatin modification, transcription activation, transcription regulation (*Yang, Froberg & Lee, 2014*). Accumulated studies have suggested that lncRNAs might be novel biomarkers for cancer

diagnosis and prognosis prediction. The prognostic value of lncRNAs have been identified in several cancer types including colorectal cancer (*Hu et al., 2014*), non-small cell lung cancer (*Zhou et al., 2015*), gastric cancer (*Zhu et al., 2016*).

In this study, we made full use of comprehensive lncRNA expression profiles derived from TANRIC database. Among these lncRNAs, one lncRNA named LINC01614 (ENSG00000230838.1) had stronger prognostic value than others. High-expression group had shorter survival than that in low-expression group. Further univariate and multivariate Cox regression analyses indicated that LINC01614 was an independent prognostic factor of BRCA. We further correlated the expression of LINC01614 with BRCA subtypes. Higher expression of LINC01614 was consistently observed in ER, PR and HER2 positive group, respectively. Additionally, Luminal A, Luminal B and HER2 subgroups had relative higher expression of LINC01614 than that in Basal-like and Normal-like subgroups. The prognostic value of LINC01614 could be observed in ER positive and HER2 negative groups, marginally in PR positive and negative subgroups, respectively. Some of the statistics were not significant probably because of the small sample size of these subgroups. When correlated the expression of LINC01614 with gene mutation status, we found the mutation status of four genes was correlated with LINC01614 expression including AOAH, CIT, HER2 and ODZ1. Higher expression of LINC01614 was observed in mutated samples. HER2 has been recognized as biomarker not only just in BRCA (*Chmielecki et al., 2015*; *Connell & Doherty, 2017*). Most of HER2 mutation patients responded to HER2 inhibitors (*Ben-Baruch et al., 2015*). AOHA was reported to have higher conditional frequencies than marginal frequencies in early time intervals. This might indicate that patients with AOAH mutation was more progressive (*Sakoparnig, Fried & Beerenwinkel, 2015*). Citron kinase was also highly expressed in BRCA and could promote cell growth and colony formation, indicating its oncogenic function in BRCA (*Meng et al., 2019*). Finally, ODZ1 was involved with increased cell proliferation and invasion in glioblastoma (*Talamillo et al., 2017*) and was identified as a biomarker in papillary thyroid carcinoma (*Cheng et al., 2017*).

Several critical pathways associated with TGF-β1 response, CDHI signaling via CTNNB1, integrin3 and integrin cell surface interaction pathways were enriched in subgroup with high expression of LINC01614. These dysregulated signaling pathways played important roles in the development and progression of cancer (*Liu & Chu, 2014*; *Seguin et al., 2015*; *Syed, 2016*). These results might imply the potential function of LINC01614 in BRCA. In terms of the prognostic value, previous studies have emphasized the prognostic value of LINC01614 in cancer. For instance, *Huang et al. (2018)* identified three lncRNAs with prognostic value in esophageal squamous cell carcinoma including RP11-366H4.1.1, LINC00460 and LINC01614. These three lncRNAs-based signature could facilitate the prognosis prediction of esophageal squamous cell carcinoma. Moreover, *Zhang et al. (2016b)* also found that LINC01614 was highly expression in ER positive BRCA which was in line with our result. These findings highlighted the potential role of LINC01614 in BRCA and it might be an useful biomarker for BRCA prognostic prediction.

 

There are several limitations for this study. Firstly, the prognostic value of LINC01614 should be further validated in cohorts with larger sample size in the future. Secondly, the differential expression of LINC01614 between BRCA and normal tissue should be validated through experiment. Thirdly, the potential role of LINC01614 in BRCA should be experimentally explored and dysregulated pathways identified in this study should also validated. Finally, as for the clinical practice, after the potential role of LINC01614 is further confirmed, the clinicians can use it to facilitate prognosis assessment of BRCA patients after surgery.

## CONCLUSIONS

This research provided an example to identify candidate lncRNAs with prognostic value in cancers. A novel lncRNA LINC01614 was identified as a potential biomarker for prognosis prediction in BRCA. This study emphasized the importance of LINC01614 and that more research should be focused on it.

## ACKNOWLEDGEMENTS

The authors are grateful for all the subjects who participated in the study.

### Funding

This work was supported by grants from the Shanghai Pudong New Area Health and Family Planning Commission Health Science and Technology Project, NO# PW2016B-5. The funders had no role in study design, data collection and analysis, decision to publish, or preparation of the manuscript.

### Grant Disclosures

The following grant information was disclosed by the authors:
Shanghai Pudong New Area Health and Family Planning Commission Health Science and Technology Project: PW2016B-5.

### Competing Interests

The authors declare that they have no competing interests.

### Author Contributions

- Yaozong Wang conceived and designed the experiments, performed the experiments, analyzed the data, contributed reagents/materials/analysis tools, prepared figures and/or tables, authored or reviewed drafts of the paper, approved the final draft.
- Baorong Song conceived and designed the experiments, performed the experiments, analyzed the data, prepared figures and/or tables, approved the final draft.
- Leilei Zhu conceived and designed the experiments, performed the experiments, contributed reagents/materials/analysis tools, prepared figures and/or tables, approved the final draft.

- Xia Zhang conceived and designed the experiments, performed the experiments, analyzed the data, contributed reagents/materials/analysis tools, prepared figures and/or tables, authored or reviewed drafts of the paper, approved the final draft.

## Data Availability

The code is available in a Supplemental File.

## Supplemental Information

Supplemental information for this article can be found online at http://dx.doi.org/10.7717/peerj.7976#supplemental-information.

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
