# Peer review of "Long non-coding RNA, LINC01614 as a potential biomarker for prognostic prediction in breast cancer"

_PeerJ, doi:10.7717/peerj.7976_

## Round 0.1 · original submission · Major Revisions

Please be sure that English will be improved and pay special attention to the statistical analysis comments of the referees.

·

Basic reporting

The authors presented the relevant information that supports the hypothesis of the study.

Experimental design

The experimental design is adequate. In addition, the authors were transplanted in providing the R programming code used in the analyzes.

Validity of the findings

The results are interesting and direct new investigations. Despite the prognostic impact well described by the authors, the pathophysiological function of LINC01614 is restricted to interpretations by GSEA that merit future investigations. However, the authors recognize these limitations in the discussion of the results.

Additional comments

Wang and colleagues report an exploratory study of potential long non-coding RNA as biomarkers and predictors of prognosis in breast cancer patients. Using a robust database, the authors identified the long non-coding RNA LINC01614 differently expressed and impacting the clinical outcome of breast cancer patients (hazard ratio> 1 and p <0.01). Interestingly, LINC01614 was an independent marker of prognosis when analyzed with classic risk covariates. The study is well written and illustrated, and R programming code was provided. This reviewer requests some clarification, as described below.

-Please include a Table describing the population of breast cancer patients included in the study (similar to the data presented in Supplementary Table 2). This information will help readers understand the population studied.

- It is unclear what the cut-off point was used for stratifying patients into high- and low-LINC01614 in analyzes in GSEA analyzes. If it was the median, I believe the correct one is "high- and low-expression" instead of "high and low-risk". Please review the use of the term in the description of results, figures, and legends.

- In Figure 2, include the number of subject in each group.

- In Figure 4, include the value of NES, FDR, and p.

Reviewer 2 ·

Basic reporting

It is very interesting work. However, there are some points that need to be addressed:

1) It is recommended that a native speaker review the paper.

2) Page 7, line 56: the abbreviation “HER2” could be replaced by sentence: “”and human epidermal growth factor receptor 2 (HER2)”.

3) In supplemental files Fig. S1. Legend use “Workflow” instead of “Work flow”

4) Results must not be interpreted within results section, but in discussion, for example: Page 11 line 143: “These results suggested that LINC01614 might contribute.....”. Check your results please.

5) Page 14, line 195: “non-samll”, to replace by “non-small”

Experimental design

1) Page 9, lines 85-92: It is not completely clear how many lncRNAS was used for each analyze (12727?, 929?, 824?, 4537?,….), is difficult to understand.

2) In supplemental files Fig. S1.: the workflow is very confusing, because there are too many tasks in the flow, authors need to reorganize the tasks flow and identify the execution order.

3) Pages 19, 20, 25 and 26: The authors in figure 3 describe a result with statistical analysis using Kaplan-Meier estimates, but this analysis was not addressed elsewhere in the manuscript. I suggest adding some information about this analysis in both material and method / results and discussion.

4) It was necessary to include in the supplemental files the tables used for statistical analysis using R.

Validity of the findings

1) Page 14, lines 206-207: the authors suggest that: “some of the statistics were not significant probably because of the small sample size”. You could have used other databases, such as: noncode, RNAcentral, lncRNAdb. It is well known that such analyzes need to be thoroughly explored.

2) Vishnubalaji et al (Cell Death Discovery, 2019, V5, N: 109, pp 109) have shown that LINC01614 as unfavorable prognostic marker in Breast Cancer Please explain why your findings are not consistent with those found by these authors.

Reviewer 3 ·

Basic reporting

The English have to be reviewed by a professional to improve the quality and clarity of the manuscript.

Experimental design

No comments.

Validity of the findings

The authors should highlight the limitations of their study.

Additional comments

This is an interesting manuscript addressing an important aspect in the field of breast cancer. The work was well designed and performed, the results are interesting and of interest and utility to workers in the same branch of the field and deserves to be published.
Tables and Figures are appropriated.

Minor comment:
- In the discussion section, the authors should highlight the limitations of their study and underline the potential application of their findings in clinics.
- The English have to be reviewed by a professional to improve the quality and clarity of the manuscript.

---

## Round 0.2 · accepted · Accept

The revised manuscript was improved and it is suitable to be published in PeerJ. I hope your paper reaches a large audience among the PeerJ readers.

·

Basic reporting

The current text has been improved. The manuscript contains interesting results, which are well contextualized and based on good references.

Experimental design

The experimental design is appropriate.

Validity of the findings

The manuscript contains interesting results and the analyzes are adequate.

Additional comments

The authors made all requested corrections. In addition, new information added improves the manuscript.

Reviewer 2 ·

Basic reporting

No comment

Experimental design

No comment

Validity of the findings

No comment

Additional comments

The authors responded appropriately to the comments.
The work is good and adds knowledge to the study of biomarkers for breast cancer prognosis.